# Who tweets climate change papers? investigating publics of research through users' descriptions

Rémi Toupin [1] *, Florence Millerand [2], Vincent Larivière[3,4]

**1** Laboratory for Communication and the Digital (LabCMO), Centre Interuniversitaire de Recherche sur la Science et la Technologie (CIRST), Université du Québec à Montréal, Montréal, Québec, Canada, **2** Laboratory for Communication and the Digital (LabCMO), Département de Communication Sociale et Publique (DCSP), Centre Interuniversitaire de Recherche sur la Science et la Technologie (CIRST), Université du Québec à Montréal, Montréal, Québec, Canada, **3** École de Bibliothéconomie et des Sciences de l'Information (EBSI), Université de Montréal, Montréal, Quebec, Canada, **4** Observatoire des Sciences et des Technologies (OST), Centre Interuniversitaire de Recherche sur la Science et la Technologie (CIRST), Université du Québec à Montréal, Montréal, Québec, Canada

\* toupin.remi@courrier.uqam.ca

**Data Availability Statement:** The code is available on Github (https://github.com/toupinr/twitterprofiles/blob/master/code_publics/20210617_PrepPublicsPropre.R) and aggregated data will be made available upon acceptance.

## Abstract

As social issues like climate change become increasingly salient, digital traces left by scholarly documents can be used to assess their reach outside of academia. Our research examine who shared climate change research papers on Twitter by looking at the expressions used in profile descriptions. We categorized users in eight categories (academia, communication, political, professional, personal, organization, bots and publishers) associated to specific expressions. Results indicate how diverse publics may be represented in the communication of scholarly documents on Twitter. Supplementing our word detection analysis with qualitative assessments of the results, we highlight how the presence of unique or multiple categorizations in textual Twitter descriptions provides evidence of the publics of research in specific contexts. Our results show a more substantial communication by academics and organizations for papers published in 2016, whereas the general public comparatively participated more in 2015. Overall, there is significant participation of publics outside of academia in the communication of climate change research articles on Twitter, although the extent to which these publics participate varies between individual papers. This means that papers circulate in specific communities which need to be assessed to understand the reach of research on social media. Furthermore, the flexibility of our method provide means for research assessment that consider the contextuality and plurality of publics involved on Twitter.

## Introduction

In recent years, Twitter became a key platform for the dissemination research [1]. As traces left by scholarly documents in tweets may reflect communication beyond traditional citations

However, full data obtained from Altmetric, Dimensions, Twitter and Web of Science require access to these services. Interested reader can contact Thomson Reuters (Web of Science: http://thomsonreuters.com/en/products-services/scholarly-scientific-research/scholarly-search-and-discovery/web-of-science.html) and Altmetric (https://www.altmetric.com/research-access/) for further access.

**Funding:** This research was funded through a SSHRC Joseph-Armand Bombardier Canada Graduate Scholarship (767-2017- 1329), the SSHRC Insight Grant Understanding the societal impact of research through social media, and received financial contribution from the CIRST. It also benefited of financial support from the Fonds de recherche du Québec – Société et Culture, the Université du Québec à Montréal, and the Canada Research Chairs Program. The funders had no role in study design, data collection and analysis, decision to publish, or preparation of the manuscript.

**Competing interests:** The authors have declared that no competing interests exist.

and in the public sphere, they were heralded as potential indicators of the so-called « societal impact of research » along with other social media metrics [2, 3]. However, the strict focus on event count (i.e. number of tweets, number of retweets) was confronted to a lack of theoretical grounding as to what these traces really measure. Scholars thus looked to investigate the contexts in which research circulate on Twitter, understood as the dimensions that give meaning to indicators [4]. Challenges remain in capturing these contextual elements as digital scholarly communication studies need to move between the scales of individual documents and aggregated corpora where contexts may shift [5]. Methodological framework also need to account that information provided on Twitter is generated by users as well as not directly organized for research purposes [6–8].

Meanwhile, discussions about issues like climate change, public health, and artificial intelligence moved to social media, highlighting the political ramifications of research [9–12]. As such, studies about science communication, policy and evaluation increasingly aim to understand the reach of scholarly outputs in the public sphere. Our study focuses on the case of climate change as representative of environmental challenges. Specifically, climate change communication aims to foster environmental action by influencing decision making and translating new knowledge in everyday practices to limit our ecological footprint [13]. Climate change issues reflect other increasingly urgent matters, such as biodiversity loss, extreme weather events, massive migrations, and scarcer access to basic resources [11, 14]. As discussions about climate change are increasingly salient on Twitter, scholars and other actors of the public diffusion of research moves to the platform to share relevant knowledge and engage with stakeholders more broadly [11, 15–17]. As some social media platforms like Twitter make their data accessible to the scholarly community, it makes it possible to directly examine the resonance of climate change research in the public sphere.

## Public conversations of climate change research on Twitter

Reflecting a large scope of topics and issues, a diversity of publics are concerned with climate change [16, 17]. On the one hand, a tracking of the release of the IPCC 5th assessment report found that the majority of Twitter engagement came from individual bloggers and concerned citizens who provided alternative framing than that of decision-makers, journalists, and scientists [17]. On the other hand, scientists discussing climate change on Twitter engage mostly with other scientists but have been seen to communicate their research to decision-makers, journalists and the general public as well [16]. Typically, scientific knowledge production and communication begins with scholars and research institutions from all disciplines, and synthesis are produced for policymakers [14]. Journalists, medias, scientists and other communication professionals then play a role communicating and framing issues in the public sphere [18, 19]. Civil society, concerned citizens, health or environment professionals, as well as political organizations and advocates also engage with climate change for personal, political or professional motivations [20, 21]. As all these actors contribute differently to discussions about climate change, the visibility of research documents on social media made the communication of related issues no longer the prerogative of scientists and journalists only [17]. The reach of climate change research may thus be modulated by the influence, background and motivationf of those who share it on Twitter.

Twitter play a significant role in informational communication as well as political discussion and action, especially for issues like climate change [16, 17, 22–24]. Within these conversations, scholars discuss relevant research with colleagues, foster new collaborations, engage in political actions, share their work more broadly, or keep in touch with the latest news [25]. This increased participation by researchers is linked with a significant volume of scholarly

documents being shared, with variations across disciplines, publication channels and cultural contexts [1, 26]. While patterns of scholarly communication on Twitter remain to be documented in detail, one key promise is that it has democratized access to research by allowing it to circulate more broadly and outside of academia. The digital traces left by scholarly documents have been heralded as potential indicators of the « societal impact of research », or altmetrics [27]. Previous research focused on the analysis of traces on Twitter as 1) data collection is easier than for most other platforms; 2) scholarly output is readily shareable through the inclusion of links in tweets; and 3) tweets are available to non-academic publics [28, 29]. However, it remains unclear what these traces reflect. On the one hand, there are multiple understandings of what is called the « societal impact of research » [3, 29–31]. On the other hand, as the communication of research on Twitter involves mediation processes and is not intended toward a clear objective, Twitter scholarly metrics do not reflect a clear phenomenon [1]. The initial focus on counts has now shifted to more comprehensive studies of the contexts in which documents are shared and what they mean for the communication of research outside of academia [4, 32]. Our study aims to further describe these contexts by focusing on the publics of climate change research as understood by their Twitter profile descriptions.

Climate change research topics range from the physical processes of climate change to its direct and indirect repercussions on communities and the environment, as well as means of mitigation, adaptation and communication to counter the ongoing process [33, 34]. Two events have been at the core of climate change research and policy discussions in recent years: the publication of the IPCC 5th Assessment Report (IPCC 5AR) in 2013 and the COP 21 leading to the Paris Agreement in 2015 [17, 35–37]. In one instance, non-elite actors–individual users who are not affiliated to a specific media, nonprofit, or scientific organization, such as bloggers, activists, or the general public–were able to draw attention, as indicated by their presence (35%) in the hundred most mentioned users, in discussing and framing the IPCC 5AR Working Group 1 contribution [17]. Scholars and research professionals contributed to both discussions–the IPCC 5AR and COP 21 events -, highlighting their interest for different topics than the general public, and indicating a shift in their public communication patterns. As such, scholars were seen to have a hybrid role of communicator and advocate, while mostly communicating with journalists and other scholars rather than directly to policymakers and the general public [16].

Conversations on Twitter build on a series of affordances, such as hashtags (#), mentions (@) or links to external documents, as well as metadata that allows for the characterization of every tweets (ex. time of publication, number of likes) and users (profile description, picture, number of followers, etc.) [16, 38, 39]. Users usually engage with accounts they are familiar with or which post content relevant to them [40, 41]. Scholarly communication on Twitter relies on the possibility of adding links to external documents to make research outputs visible [39]. Authors and publishers may tweet links to their papers to promote them, and eventually foster engagement that will benefit them–such as an higher number of citations–whereas scholars may tweet or retweet documents they find relevant [3]. Altogether, users who actively engage with a paper may do so by being prompted by other accounts, whether because a publication was relevant, funny or controversial [42, 43]. Influential users, such as communicators or celebrities, may engage their network more easily, while communities may be created around specific documents to topics [4, 44].

## Investigating and representing users in public communication of research on Twitter

On Twitter, the abundance of informational content fostered an engagement by political users, communication professionals and organizations, as well as a representation of actors of the

knowledge economy [16, 45]. An account may represent an individual, a project, an organization, or a feed of content [46, 47]. The demographics of Twitter reflect those of the high-middle class of the population, mostly white young professionals, although these demographics changes as we scale down to specific communities [48]. As for scholars, doctoral students and young researchers tend to be at the forefront, while some disciplines, mostly those dealing with social issues such as health sciences, economics or social sciences, are more visible [3, 49]. Our study focuses on the analysis of words and expressions in Twitter user bios as a proxy of who tweet climate change research papers since assessing who share scholarly documents at a larger scale entails reducing identity to specific markers [28, 46]. We hypothesize that the expressions employed by users to describe themselves act as identity markers through which they engage with other users [50]. Building on previous work on the identification of users sharing scholarly documents on Twitter, we identified 8 relevant categories of markers–academia, communication, political, professionnal, personal, organizations, publishers, and bots–to classify who tweet climate change research papers [5, 16, 28, 51].

Methods that capture who engage with scholarly documents on Twitter usually rely on automatic textual analysis of Twitter bios [16, 28, 51–53] or manual coding [41, 54]. Altmetric also provides an identification of users in its database in four categories distinguishing between researchers, science communicators, practitioners and general public [55]. However, their approach has limitations as it encompasses the "general public" as all the users who do not match to the first three categories. Other approaches relied on the characterization of the social network by which documents flow [16, 32, 54, 56]. Usually employed in conjuncture with textual analysis of Twitter profile description, these methods aim to understand how discussions or communities build up around scholarly documents. More direct approaches rely on the matching of bibliometric information with Twitter data to capture the scholars involved on Twitter [57], as well as the use of Twitter lists [51, 58]. Our method build on these by investigating the categories of users sharing climate change research papers on Twitter through specific expressions in Twitter profiles descriptions. As such, we account for the multiple identity markers used in a description to further assess the complexity through which someone may engage with research documents. We did not take into account the order in which these markers appear as we wanted to have a general overview of how users present themselves without apposing any judgement on which identities are more important.

## Purpose of the study

Profile description is often the primary information through which we assess someone else identity on Twitter [28]. As such, it is useful to assess someone else inclination toward specific topics on Twitter. Twitter bios also are a widely used proxy in informetrics studies to determine who are the users engaging in scholarly communication on Twitter [16, 28, 51–53]. As such, our study examines a categorization of users by analyzing specific keywords in Twitter descriptions.

Our main objective is to look how much research papers about climate change issues permeate outside of academia by examining the specific categories of users sharing said papers. As such, we focus our analysis on general categories of "markers" about users who may have an interest toward such research. We classify the descriptions of accounts who shared at least one link to a climate change research paper by linking them to the expressions collated for each category. Thus, we aim to assess the reach of scholarly documents in specific categories of users involved in the public communication of climate change research on Twitter:

- RQ: Who share climate change research papers outside of academia?

Methodologically, word detection highlights which expressions users provide in their Twitter descriptions. It contributes to our understanding of scholarly communication on social media by assessing how different identity markers may be used within single bios and the communities linked to these expressions. However, it elicits methodological discussions as our approach does not aim to attach unique identity markers to accounts, but rather highlights the multiple ways through which users may express themselves on Twitter. Our paper addresses these considerations by investigating who share climate change research articles through a word detection method of Twitter bios. Thus, we aim to provide insights on how users present themselves to others in climate change research discussions, especially outside of academia.

## Material and methods

### Data collection and Twitter metrics

For this study, we built a dataset of 2015 and 2016 research articles (n = 4 730) indexed in the Web of Science (WoS) of Clarivate Analytics through the internal database of the Observatoire des sciences et des technologies (OST). We then collected tweets containing a link to these papers as well as information about the users who published these tweets by cross-referencing the information gathered from WoS with the Altmetric–a division of Digital Science (Springer) tracking scholarly documents on social media–database via the Digital Object Identifier (DOI). We accessed the database through an October 2017 copy provided to the Observatoire des sciences et des technologies (OST). We used data from the WoS database as it indexes a large number of research documents from several fields as well as extensive bibliometric information about said research documents [59]. However, attention is directed to a specific set of scientific literature published in English [60]. To select relevant papers about climate change research, we focused on those published in 2015 in 2016 that included the keywords "climate change", "global warming" or "IPCC" in the title and for which a DOI (*Digital Object Identifier)*, a unique identifier referencing online documents was provided. We chose the years as the Paris Agreement, approved on 12 December 2015, marks a critical juncture for science communication and climate change engagement [35, 37]. Since Altmetric information was provided through a data dump in October 2017, they also were the latest years for which we had coverage of all the research articles through both years at the time of data collection in September 2018. We focused on the title as it is a direct metadata to assess a paper relevance [39]. It is also the information that appears the most in tweets sharing a link to a paper. This query does not retrieve all publications in climate change research; rather, we wanted to collect a set of papers directly related to climate change. As such, our aim is not to provide an extensive analysis of the field, as is done elsewhere [33, 61]. Data collected includes the paper DOI, title, abstract, name of first author, journal of publication, NSF discipline and specialty, number of pages, number of references, number of authors, number of citations, number of tweets, number of accounts, time of first and last tweet, and tweetspan. Collected tweets metadata include paper DOI, tweet author ID, tweet ID, tweet content, time of publication and retweet order. User information include, at the time of tweet, the author ID, author name, account description, account URL, geographic stamp, number of followers, number of papers tweeted, number of tweets, time of first and last tweet, and tweetspan. Queries to the WoS and Altmetric database were made in SQL and we exported the results for further access. Data collection complied with the terms and conditions of WoS, Altmetric and Twitter through data providing agreements with the Observatoire des sciences et des technologies (OST).

Following recommendations from previous studies, we computed several Twitter metrics to further describe our dataset for tweet activity [1, 4]. Computed metrics include the number of papers tweeted, number of tweets, Twitter coverage (i.e., % of tweeted papers), Twitter

density (i.e., number of tweets per paper) and intensity (i.e., number of tweets per tweeted paper), number of users, user density (number of users per paper) and intensity (i.e., number of users per tweeted paper), number of papers retweeted, retweet coverage (i.e.,% of retweeted papers), share of retweets (i.e., % of tweets that are retweets), retweet density (i.e., number of retweets per paper), and retweet intensity (i.e. number of retweets per tweeted document). These metrics further characterize our dataset by providing an assessment of Twitter engagement across scholarly and social media objects. We computed all metrics by uploading our dataset in an R dataframe and handling it using the *tidyverse* package as well as basic calculation functions [62]. Plots were created using the *ggplot2* package [63].

Our dataset includes 2 376 papers published in 2015 and 2 354 in 2016. The papers were published in 1 062 journals, from which 46 have published more than 20. The journals *Climatic Change* (n = 178), *PLOS ONE* (n = 139), *Global Change Biology* (n = 97), *Regional Environmental Change* (n = 70), *Scientific Reports* (n = 63), *Environmental Research Letters* (n = 55), *Journal of Climate* (n = 54) and *Nature Climate Change* (n = 51) each have published more than 50 articles. This illustrates how climate change research is a broad field encompassing various disciplines, as well as showing the diverse possibilities of publication whether in specialized or more general journals. We collected information from 41 108 tweets–among which 23 831 were retweets–sent by 21 844 unique accounts linking to 2 628 papers. The 56% Twitter coverage of our dataset is comparable to that of medical and health research and average of more than eight users sharing tweeted papers indicate significant engagement toward the papers gathered in our study (1). Among tweeted papers, 1 961 were shared by at least two users, 667 by more than ten, 338 by more than twenty, 129 by more than fifty, and 47 by over than a hundred user. Also, 1 319 and 1 308 papers published respectively in 2015 and 2016 were shared at least once, for 21 985 and 19 349 tweets by 12 815 and 11 461 unique users. Tweeted papers were published in 646 journals, 16 of which published more than 20 papers. *Climatic Change*, *PLOS ONE* and *Global Change Biology* published most tweeted papers, whereas *Nature Climate Change*, *Science* and *PNAS* account for the three journals publishing the most papers tweeted by more than a hundred users (Fig 1).

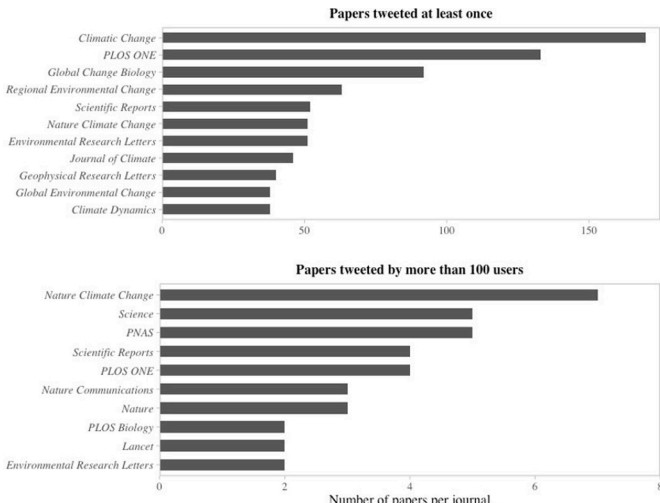

**Fig 1. Distribution of climate change research tweeted papers in scientific journals.** Depicted in the above histogram are the ten journals that published the most tweeted papers in our dataset, and below are the ten journals that published the most papers tweeted by more than 100 users.

## Textual analysis of Twitter profile descriptions

To understand how climate change research permeate outside of academia, we focused on the textual analysis of these descriptions. Specifically, we looked at specific expressions that indicate how users present themselves on Twitter [28]. As such, our analysis does not aim to provide an exact mapping of the scientists or journalists on Twitter [57, 64]. Rather, we understand expressions in Twitter descriptions as a proxy to investigate the potential reach of scholarly documents outside of academia. We looked at Twitter descriptions using a dictionary of expressions for eight relevant categories of identity markers built on previous research [5, 16, 28, 51] (Table 1). We built a first version of the codebook by manually coding a sample of a thousand descriptions, and we then improved it through several iterations of the analysis, running the code, comparing with our manual coding, and then modifying the codebook accordingly. We removed all accounts with no descriptions (NULL), lowered all cases, and removed numbers, URLS, emojis, stopwords and punctuation signs except for the hash (#) and at signs (@) from all remaining descriptions. Our inputted data frame featured one user per paper per row, and we removed duplicates as we filtered down our analysis. We assigned expressions to categories using the *tidyverse* packages and then matching them with corresponding words in the descriptions [62]. Our method may assign more than one category to a description, and thus considers the possibility for someone to provide multiple identity markers. We completed our analysis by looking at the table of descriptions sorted in different categories to assess the potential representations of users mobilizing specific words. We organized our specific observations by looking at the coverage of these categories in 10 highly tweeted papers. Thus, we describe how well these categories may be investigated in studies about the communication of research on Twitter, specifically climate change research. We provided a category to 70% of Twitter bios for this study. Unidentified descriptions include those not written in English or French–language for which we were sufficiently fluent–and those not specific enough to be matched to one of our categories.

**Table 1. Categories and matching expressions used for textual analysis of Twitter descriptions.**

| Categories | Ex. of specific expressions | Ex. of Twitter descriptions |
|---|---|---|
| Academia | researcher, professor, phd, biologist, postdoc | Post-doctoral coastal scientist / engineer @unisouthampton, UK. Researches #sealevelrise #impacts #adaptation #islands #deltas. Also likes #cows. |
| Personal | yoga, music, father, mother, cat | Curious, Mother of two, Retired. |
| Professional | physician, manager, engineer, strategist, veterinarian | Environmental attorney. Climate change terrifies me. |
| Political | advocate, policy, councillor, social justice, #standupforscience | Mayor of @CityKitchener. Community promoter of Kitchener & @WRAwesome-ness. Past Prez of @FCM_online. Treasurer of @uclg_org. Motto: Live ~ Love ~ Laugh |
| Communication | journalist, writer, author, podcast, youtuber | Journaliste, directrice de la rédaction de @Sante_Magazine. Mes tweets n'engagent que moi. Compte perso |
| Organization | university, institue, media, association, research centre | Updates from AAAS, the American Association for the Advancement of Science. Open minds. Join us. http://tinyurl.com/JoinAAAS |
| Publishers | Wiley, Sage, Elsevier, issn, journal | Published by Oxford University Press, AoB PLANTS features peer-reviewed articles on all aspects of environmental and evolutionary plant biology. |
| Bots | bots, RSS, paper alerts, retweets from, daily updates | A Bots tweeting new research from the Canadian Government (NRC, AAFC, EC, DFO & NRCan). Not affiliated the Government of Canada |
| Unassigned | | An unknown particle in this Universe |

The above table presents the categories used in our study with a selection of five correspondings expressions and an example of Twitter profile descriptions. The complete list of expressions can be found at https://doi.org/10.6084/m9.figshare.8236598.v3 and https://github.com/toupinr/twitterprofiles/blob/master/code_publics/20210617_PrepPublicsPropre.R [65].

## Results

### General results

Our final dataset included 19 783 unique Twitter accounts. We assigned at least one category to 69.9% of the accounts (n = 13 821) by using our code to detect expressions in Twitter bios, and 36.2% (n = 7 155) matched to only one category, and 33.7% (n = 6 666) to multiple categories (Table 2). Academia is the largest category across all papers at 5 545 users, representing 28% of our dataset. Personal assignations represent a quarter of the dataset. Publishers and bots are the less visible categories, both under 2%. However, accounts posting automated content do not usually identify as such. Therefore, a low representation of automated accounts is mostly an artefact of the method used in our study, as it is probably much higher (24). The number of unique users was slightly higher in 2015 (n = 11 745) than 2016 (n = 10 467). We also notice a higher uptake by Personal, Professional, Political and Communication public in 2015, potentially indicating higher engagement outside of academia. Academia, Organizations, Publishers and Robots are proportionally more represented in 2016. Academia and Organizations engaged with the most papers in our dataset, whereas Political, Publishers and Bots assigned users engaged with the least. Bots and Academia have the lowest median in number of followers, Communication and Publishers the highest. This may indicate that communicators and publishers tend to fulfill a role of sharing research with a larger network of persons than other group of users. These trends are similar for 2015 and 2016, although number of followers of users tweeting the papers of our dataset were higher in 2015.

Looking at overlaps between categories, Academia/Personal and Academia/Organizations are the most present in users' descriptions (Table 3). The large number of overlaps across all categories indicate that the assignations of based on unique identity markers significantly reduces the complex ways in which users present themselves to others on Twitter. As such, the complexity of identifying who share research papers on Twitter may be best documented by looking at multiple categories and in the context of individual papers.

**Table 2. Summary of results across all papers.**

| | All papers | | | | 2015 | | | | 2016 | | | |
|---|---|---|---|---|---|---|---|---|---|---|---|---|
| Type of publics | Total n of users | % w unique assignations | N of papers | Median n of followers | Total n of users | % w unique assignations | N of papers | Median n of followers | Total n of users | % w unique assignations | N of papers | Median n of followers |
| Academia | 5 545 | 32.2% | 1 613 | 502 | 3 193 | 31.1% | 816 | 603 | 3 343 | 34.4% | 797 | 504 |
| Personal | 4 939 | 32.0% | 1 460 | 655 | 3 071 | 32.2% | 760 | 757 | 2 387 | 29.6% | 700 | 636 |
| Professional | 2 963 | 24.0% | 1 045 | 819 | 1 782 | 22.1% | 527 | 950 | 1 506 | 24.4% | 518 | 790 |
| Political | 2 564 | 28.0% | 921 | 863 | 1 651 | 27.0% | 469 | 990 | 1 240 | 28.0% | 452 | 833 |
| Communication | 2 237 | 26.2% | 1 020 | 1 128 | 1 393 | 25.9% | 505 | 1245 | 1 140 | 25.9% | 515 | 1 111 |
| Organization | 4 201 | 37.6% | 1 656 | 726 | 2 462 | 38.1% | 836 | 870 | 2 378 | 37.0% | 820 | 701 |
| Publishers | 357 | 38.1% | 650 | 1 499 | 213 | 38.5% | 327 | 1778 | 217 | 39.2% | 323 | 1 688 |
| Bots | 101 | 61.4% | 466 | 440 | 61 | 65.6% | 202 | 482 | 68 | 38.8% | 264 | 592 |
| Unassigned | 5 962 | | 1 680 | 632 | 3 457 | | 859 | 724 | 3033 | | 821 | 616 |
| Total | 19 783 | 36.2% | | | 11 745 | 36.1% | | | 10 467 | 37.0% | | |

The above table presents the absolute number of profiles assigned to each category (Total n of users), the% of profiles with only one assignation for each category (% w unique assignations), the number of papers tweeted by at least one user per category (N of papers), and the median number of followers of the users assigned to each category (Media n of followers). Results are presented for the whole dataset investigated in this study as well as splited between years.

**Table 3. Number of Twitter bios with unique of multiple categories.**

| Type of publics | Bots | Publishers | Organization | Communication | Political | Professional | Personal | Academia |
|---|---|---|---|---|---|---|---|---|
| Academia | 3 | 48 | 1 528 | 725 | 656 | 870 | 1 628 | 1 786 |
| Personal | 13 | 35 | 709 | 772 | 852 | 1 017 | 1 579 | |
| Professional | 4 | 27 | 674 | 478 | 554 | 710 | | |
| Political | 3 | 29 | 474 | 317 | 718 | | | |
| Communication | 12 | 55 | 318 | 586 | | | | |
| Organization | 6 | 110 | 1 578 | | | | | |
| Publishers | 8 | 136 | | | | | | |
| Bots | 62 | | | | | | | |

The above table presents the absolute number of dual overlaps in individual Twitter profile descriptions per category. Cells in blue present the number of profiles assigned to only one category. Cells at the intersection of two categories present the number of Twitter profiles assigned to both categories.

## Publics in highly tweeted papers

We looked at the five most tweeted papers per year to categorize who are users sharing popular scholarly documents about climate change research on Twitter (Table 4). Most papers have between 65 and 75% of the users sharing them assigned to at least one category, except the article *Oxygen isotope in archaeological bioapatites from India: Implications to climate change and decline of Bronze Age Harappan civilization* at 46.6%. The representation of users assigned to Academia is lower than the mean of 28% of our dataset for nine out of the ten. Overall, variations between how categories are represented between articles highlight the different contexts in which individual papers are shared, thus providing a basis to assess how and why they get attention.

The *Ecological networks are more sensitive to plant than to animal extinction under climate change* was shared by 47.8% of the accounts assigned to Academia, indicating a significant engagement by the research community. For some papers, the Twitter profiles of users assigned to Academia have important overlap with other categories (S1 Table in S1 File). For example, the paper *Health and climate change: policy responses to protect public health* was tweeted by 27.8% of the Academia assignations overlapping with Professional category. A manual validation of the results indicates that most assignations to Academia indeed represent scholars and researchers, with very few discrepancies. This highlights the potential of word detection to assess the representation of scholars in a dataset of papers shared on Twitter, at least in terms of precision. However, there is a possibility that scholars were not categorized as such depending on the words used in their Twitter bios. Other methods are better suited to assess overall participation by scholars [3, 57, 58], whereas our method focuses on their participation according to other groups of users.

The proportion of users assigned to the Communication category is higher when more users shared a specific paper. Some papers show important overlaps between Communication and Academia such as *Ecological networks are more sensitive to plant than to animal extinction under climate change* (56%) and *Assessing the Performance of EU Nature Legislation in Protecting Target Bird Species in an Era of Climate Change* (44.4%) (S2 Table in S1 File). There is also large variations for the Communication and Political overlaps, ranging from 2.6% (*Climate change impacts on bumblebees converge across continents*) to 23.9% (*Accelerating extinction risk from climate change*), and the Communication and Professional overlaps, ranging from 12.5% (*Global and regional health effects of future food production under climate change: a modelling study*) to 34.2% (*Climate change impacts on bumblebees converge across continents*). The Personal category has the most overlap with Communication in the most tweeted papers, with no

**Table 4.  General results of the word detection analysis.**

| | | | | Acad | Perso | Pro | Pol | Comm | Org | Pub | Bots | |
|---|---|---|---|---|---|---|---|---|---|---|---|---|
| Title | Journal | Year | N of users | % of users | % of users | % of users | % of users | % of users | % of users | % of users | % of users | % unassigned |
| **Total** | | | 19 783 | 28.0% | 25.0% | 15.0% | 13.0% | 11.2% | 21.2% | 1.8% | 0.5% | 30.1% |
| *Climate change in the Fertile Crescent and implications of the recent Syrian drought* [66] | PNAS | 2015 | 1 760 | 13.9% | 34.7% | 12.2% | 19.1% | 14.8% | 10.5% | 0.7% | 0.2% | 36.1% |
| *The geographical distribution of fossil fuels unused when limiting global warming to 2 degrees C* [67] | Nature | 2015 | 1 265 | 23.1% | 27.7% | 19.4% | 23.8% | 13.0% | 20.2% | 0.6% | 0.3% | 25.1% |
| *Accelerating extinction risk from climate change* [68] | Science | 2015 | 749 | 20.3% | 31.5% | 12.7% | 18.4% | 12.3% | 13.5% | 0.1% | 0.3% | 34.8% |
| *Health and climate change: policy responses to protect public health* [69] | Lancet | 2015 | 481 | 26.2% | 30.4% | 23.1% | 19.8% | 11.4% | 27.4% | 1.2% | 0.2% | 21.8% |
| *Climate change impacts on bumblebees converge across continents* [70] | Science | 2015 | 337 | 27.0% | 30.0% | 17.5% | 11.0% | 11.3% | 24.0% | 1.2% | 0.0% | 27.9% |
| *Analysis and valuation of the health and climate change cobenefits of dietary change* [71] | PNAS | 2016 | 659 | 25.2% | 30.3% | 17.6% | 20.2% | 12.7% | 16.4% | 1.2% | 0.0% | 29.3% |
| *Oxygen isotope in archaeological bioapatites from India: Implications to climate change and decline of Bronze Age Harappan civilization* [72] | Scientific Reports | 2016 | 537 | 10.4% | 21.2% | 15.6% | 8.2% | 9.3% | 5.6% | 0.4% | 1.5% | 53.4% |
| *Global and regional health effects of future food production under climate change: a modelling study* [73] | Lancet | 2016 | 347 | 21.6% | 28.8% | 18.2% | 16.1% | 9.2% | 22.2% | 2.0% | 0.0% | 28.8% |
| *Ecological networks are more sensitive to plant than to animal extinction under climate change* [74] | Nature Communications | 2016 | 276 | 47.8% | 21.4% | 10.9% | 9.1% | 9.1% | 26.1% | 1.4% | 0.0% | 25.4% |
| *Assessing the Performance of EU Nature Legislation in Protecting Target Bird Species in an Era of Climate Change* [75] | Conservation Letters | 2016 | 238 | 26.1% | 32.4% | 18.9% | 13.0% | 7.6% | 22.3% | 0.8% | 0.0% | 26.5% |

The above table presents a summary of the results of the word detection analysis on the whole dataset and the 5 most tweeted papers of 2015 and 2016. Columns ranging from Acad to Bots (Acad = Academia; Perso = Personal; Pro = Professional; Pol = Political; Comm = Communication; Org = Organization; Pub = Publishers) represent the percentage of Twitter bios assigned to each category according to the number of users (N of users). The last column indicate the percentage of Twitter bios not assigned to any category.

paper having a proportion lower than 32% and the highest at 52.5% (*Accelerating extinction risk from climate change*). This indicate that communicators may use a large share of personal keywords and expressions to build their perceived identity on Twitter.

Assignations to the Political category range from 8.2% (*Oxygen isotope in archaeological bioapatites from India*: *Implications to climate change and decline of Bronze Age Harappan civilization)* to 23.9%. (*The geographical distribution of fossil fuels unused when limiting global warming to 2 degrees C*). Papers focusing on sensitive topics (such as *The geographical distribution of fossil fuels unused when limiting global warming to 2 degrees C* and *Health and climate change*: *policy responses to protect public health*) may engage more users with significant political motivations. Two papers have an elevated overlap between Academia and Political (*Ecological networks are more sensitive to plant than to animal extinction under climate change*; 36% and *Assessing the Performance of EU Nature Legislation in Protecting Target Bird Species in an Era of Climate Change*; 32.3%) (S3 Table in S1 File). This may indicate that a significant share of users from the research community also embrace political action when it comes to climate change.

Users assigned to the Professional category range from 10.9% (*Ecological networks are more sensitive to plant than to animal extinction under climate change*) to 23.1%. (*Health and climate*

*change*: *policy responses to protect public health*). The paper *Oxygen isotope in archaeological bioapatites from India*: *Implications to climate change and decline of Bronze Age Harappan civilization* has close to half (45.2%) of its Professional assignations not overlapping with any other categories (S4 Table in S1 File). Two papers, *Global and regional health effects of future food production under climate change*: *a modelling study* and *Assessing the Performance of EU Nature Legislation in Protecting Target Bird Species in an Era of Climate Change*, have a low share of overlap with Communication at respectively 6.3% and 8.9%. However, this paper has a large share of overlap with the Personal category at 57.8%. Overall, Professional and Personal overlapping is high across all the most tweeted papers, with the lowest at 29.8%.

The largest share of Personal assignations is with the most tweeted paper, *Climate change in the Fertile Crescent and implications in the recent Syrian drought*, at 34.7%, whereas two papers (*Oxygen isotope in archaeological bioapatites from India*: *Implications to climate change and decline of Bronze Age Harappan civilization* at 21.2% and *Ecological networks are more sensitive to plant than to animal extinction under climate change* at 21.4%) have the lowest share. These two papers represent both extremes of the range of unique assignations, at respectively 47.4% and 18.6% of accounts assigned to the Personal category (S5 Table in S1 File). The second most important share of unique assignations is with the paper *Climate change in the Fertile Crescent and implications in the recent Syrian drought* at 42.9%. The paper *Ecological networks are more sensitive to plant than to animal extinction under climate change* show here again a high overlap of Personal assignations with the Academia category, at 59.3%. Finally, the paper *Assessing the Performance of EU Nature Legislation in Protecting Target Bird Species in an Era of Climate Change* has a low overlap of Personal assignations with the Communication category (7.8%) while high with the Professional category (33.8%). These results highlight the significant use of personal identity markers across all publics. In this regard, users assigned only to the Personal category may represent what is coined as the lay public.

## Discussion

A key challenge in assessing who tweets scholarly documents through Twitter profile descriptions is defining the use and meaning of identity markers in expressions employed by users. Analysis relying only on Twitter data is a complex endeavor as we seldom know who is exactly behind an account. To circumvent some of these issues, our analysis categorized profile according to eight types of users sharing climate change research papers on Twitter. Specifically, we categorized expressions and keywords used in Twitter profile descriptions to assess how they represent identity markers, and so type of users sharing climate change research papers. As with other studies, the detection of words related to the academic world is precise in that it reflects potential individual users involved in research, although it does not distinguish how close their research interests are to the topic at hand [57, 76]. The overlap with other categories also shows how actors of research are not restricted to this role, whether through communicational (science communication), professional (administrative functions), political (policy making) or simply personal (being a parent, having pets, hobbies) activities [77]. Profiles categorized in Communication mostly encompasses journalists and communication professionals, authors, artists, and overlaps with political and professional expressions represents users who may engage in political campaigning, policymaking. Political assignations highlight users who present themselves through social issues and activism, some through related professional work. Professional assignations highlight those whose work relate closely to climate change mitigation efforts, for example risk management, or other specific professional activities, such as veterinarians or lawyers. Finally, personal assignations indicate how users identify themselves through their hobbies or personal interests and relationships. When

it is the only categorization, it may indicate users who engage with research through pure curiosity, thus incarnate what we commonly refer to as the "general public" [46, 78].

Assignations to organizations represent both organizational accounts (universities, departments, centers, governmental institutions, private companies, etc.) and individuals who employ expressions relating to these institutions [53]. Thus, the proportion of assignations to Organizations represent both these organizations through their specific accounts as well as individuals who use these organizations as identity markers. However, the Organizations category relies on words that have meanings beyond clear groups or institutions, such as "society". A word like society may express a delineated entity, such as the "Society for X Research", or an abstract entity, such as society as the realm of social interactions. Future research needs to take this into account, whether by adding to the dictionary, excluding problematic expressions, or refining interpretations, depending on the goals at hand.

The relative frequencies of assignations within articles provide an overall assessment of the potential groups who shared research documents on Twitter. Focusing on a selection of highly tweeted papers in climate change research in 2015 and 2016, our results indicate that expressions relating to Academia and Personal identity markers are used to a large extent, whether through unique assignations or overlap other categories. The academic community is usually the largest group sharing research on Twitter, with a diversity of different publics across individual papers. Meanwhile, political assignations look to be more present in papers discussing sensitive topics, such as fossil fuel consumption or the contribution of climate change to geopolitical conflicts. Knowing how various publics engage with individual papers may help detect trends in public communication of research, both in general datasets or specific subsets. It may also help to document how papers circulate within specific communities. For example, we see quite different patterns between the papers *Ecological networks are more sensitive to plant than to animal extinction under climate change* and *Oxygen isotope in archaeological bioapatites from India*: *Implications to climate change and decline of Bronze Age Harappan civilization*. The first one has a significant share of users assigned to Academia, hereby indicating a significant participation by the research community. The other paper has the lowest share of academic participation of the papers showcased in this study. These examples illustrate how estimates of the variety of users sharing individual papers provide a basis to further contextualize the repercussions of research dissemination on Twitter, especially for topics like climate change that resonate differently across distinct groups such as conservation specialists or concerned citizens. A qualitative survey of the results then provides meaning to these estimates and helps validate and refine each category.

Our study has some limits due to the mediated characteristics of Twitter data, and the epistemological and technical choices made prior and through the design of the study. Word detection works as a proxy for the categorization of users but does not provide a direct assessment of "who" really participates to discussions about climate change research on Twitter [1, 28]. It is hardly possible to access the user behind an account through automated data analysis. We also rely on information chosen explicitly by the user and do not have access to all the choices made for the preparation of their Twitter profile description. While, this allows users to identify themselves to others in their own words and make their identities visible in ways they chose, our interpretations are based solely the identify markers we have access to.

We also chose to categorize users Twitter profile by assigning categories to them and by going back and forth between an automatic method of expressions detection and a manual coding. These categories were chosen according the litterature to make sense of public research communication activities on Twitter [1, 3, 16, 28, 51]. We hypothesized that examining the words and expressions employed by users would allow to investigate the extent to which various communities share climate change research on Twitter. Our analysis relies

mainly on the presence or absence of specific expressions. This allows for flexibility to assess who tweets different sets of documents as context is dynamic. However, choices need to be made clear to provide interpretation about how specific expressions are used in various contexts. Moreover, some categories may need to be revisited for further development or be assessed in conjunction with other methods, such as distinguishing between individuals or organizations or detecting accounts posting automated content [46].

Despite these caveats, this study present elements to assess the potential publics of climate change research on Twitter by taking context into account. We see how individual documents are shared beyond strict scholarly communities and in specific groups. Our results, while focusing on highly tweeted papers about climate change, indicate that academia is the main group involved, but that specific papers also reach a variety of publics, whether professional, political and personal, depending on the context in which they are shared. The method we deployed is readily usable across large sets of documents, flexible in that words and categories may be modulated and refined according to research objectives, and provide key insights about 'who' tweets research on Twitter. It may also be used in conjunction with other methods to further describe these assessments as well as provide statistical or qualitative observations to what is observed. By focusing on the Twitter descriptions of users, we can work directly with identity markers chosen by users on Twitter. Future research may refine the choices made building this tool. Overall, it serves as a step for future work about who tweets research document in conjonction with other methods, such as social network analysis [32]. It also provides new elements to contextualize the reach of scholarly documents on Twitter.

## Conclusion

This study focused on the categorization of users sharing climate change research papers on Twitter by using a word detection method based on profile descriptions. While our results do not provide a direct assessment of who tweet research due to the characteristics of identity markers in Twitter profile descriptions, it provides insights about how documents may permeate outside of academia and in various communities. Focusing on a subset of highly tweeted papers about climate change, we see how different group of users share research papers on Twitter. As such, we provide information about who tweets individual documents to further describe the specific contexts in which this research circulates. We proposed a framework that is flexible as we presented one set of categories and expression. These may be easily changed, based on qualitative assessment, to assess different group of users in other Twitter scholarly communication research. Moreover, moving between automated word analysis and qualitative assessment helps inform the interpretations of who is represented through these groups. As such, it highlights how contextual observations will help to better inform the reach of research documents on social media.

## Supporting information

**S1 File.**
(ZIP)

## Acknowledgments

We would like to thank Stefanie Haustein and Juan Pablo Alperin from the ScholCommLab for their help and feedback regarding the analysis. We would also like to thank Matisse Dagenais and Sandrine Dagenais in helping build the codebook.

## Author Contributions

**Conceptualization:** Rémi Toupin, Florence Millerand, Vincent Larivière.

**Data curation:** Rémi Toupin.

**Formal analysis:** Rémi Toupin.

**Funding acquisition:** Rémi Toupin, Florence Millerand, Vincent Larivière.

**Investigation:** Rémi Toupin.

**Methodology:** Rémi Toupin.

**Project administration:** Rémi Toupin, Florence Millerand, Vincent Larivière.

**Resources:** Florence Millerand, Vincent Larivière.

**Supervision:** Florence Millerand, Vincent Larivière.

**Validation:** Rémi Toupin, Florence Millerand, Vincent Larivière.

**Visualization:** Rémi Toupin.

**Writing – original draft:** Rémi Toupin.

**Writing – review & editing:** Rémi Toupin, Florence Millerand, Vincent Larivière.

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
