## [Decision Letter · Decision Letter 0]

23 Sep 2021

PONE-D-21-17099Who tweets climate change papers? Investigating publics of research through users’ descriptionsPLOS ONE

Dear Dr. Toupin,

Thank you for submitting your manuscript to PLOS ONE. After careful consideration, we feel that it has merit but does not fully meet PLOS ONE’s publication criteria as it currently stands. Therefore, we invite you to submit a revised version of the manuscript that addresses the points raised during the review process.

Please take into consideration all Reviewers comments. However, I would like to emphasize few key issues

1. Clear language, clearly defined terms and claims as pointed out by R#1, R#2 and R#3. Please either use different wordings or generally accepted definisions (backed up by proper references). If they do not exist, please make sure that they are defined in the manuscript.

2. Please add missing references which would support the statements and assumptions in the manuscript, as well as provide a complete picture of previous research on the topic of the manuscript(R#1, R#2)

3. Please ensure that conclusions are justified by the results (R#1, R#2)

4. Please ensure that the paper has a clear research objective (R#2)

5. Please make sure that the data availability is in line with  PLOS Data policy (R#3)

We look forward to receiving your revised manuscript.

Kind regards,

Piotr Bródka

Academic Editor

PLOS ONE

Journal Requirements:

2. In your Methods section, please include additional information about your dataset and ensure that you have included a statement specifying whether the collection method complied with the terms and conditions for the websites from which you have collected data.

“We would like to thank Stefanie Haustein and Juan Pablo Alperin from the ScholCommLab for their help and feedback regarding the analysis. We would also like to thank Matisse Dagenais and Sandrine Dagenais in helping build the codebook. This research was funded through a SSHRC Joseph-Armand Bombardier Canada Graduate Scholarship (767-2017-1329), the SSHRC Insight Grant *Understanding the societal impact of research through social media*, and received financial contribution from the CIRST.”

“This study was funded by the Social Sciences and Humanities Research Council of Canada, the Fonds de recherche du Québec - Société et Culture, the Centre interuniversitaire de recherche sur la science et la technologie, the Université du Québec à Montréal and the Canada Research Chairs program. The funders had no role in study design, data collection and analysis, decision to publish, or preparation of the manuscript.”

Reviewers' comments:

Reviewer's Responses to Questions

**Comments to the Author**

1. Is the manuscript technically sound, and do the data support the conclusions?

Reviewer #1: Partly

Reviewer #2: Partly

Reviewer #3: Yes

2. Has the statistical analysis been performed appropriately and rigorously? 

Reviewer #1: Yes

Reviewer #2: Yes

Reviewer #3: N/A

3. Have the authors made all data underlying the findings in their manuscript fully available?

Reviewer #1: Yes

Reviewer #2: Yes

Reviewer #3: No

4. Is the manuscript presented in an intelligible fashion and written in standard English?

Reviewer #1: Yes

Reviewer #2: Yes

Reviewer #3: Yes

5. Review Comments to the Author

Reviewer #1: This is a very interesting manuscript that touches upon an important topic (climate change) and employs a viable method (iterative textual analysis) to analyse Twitter data. Yet, while I see the potential of this approach and would highly like to encourage to continue and further refine this, there are some issues I have encountered that should be addressed in order to further enhance the quality of the manuscript

Introduction

- Please elaborate on what you mean by “messy data from Twitter” (lines 53-55)

- “As such, science communication and policy increasingly aim to understand the resonance of research in the public sphere.” (lines 58-59) – Really? Please provide more information on this and elaborate why you think this is the case. More specifically, policy has often been criticized for detaching itself from research and rather follow “popular opinion”.

Public conversations of climate change research on Twitter

- The cited articles (lines 71-82) seem to fit the argument. However, more information is needed on the specific studies, in order to better understand and contextualize their importance for this manuscript.

- “While patterns of scholarly communication on Twitter remain to be documented in detail, it is considered to have democratized access to research.” (lines 91-92) – this is a very bold statement and should be supported by further references. More specifically, one could argue that OER and OpenAccess publications have contributed more to this suggested process than Twitter.

- The authors refer to Haustein, as well as Diaz-Faes – this needs more information and elaboration as it appears to be key to the manuscript.

- What exactly are “non-elite” actors? (line 121)

Investigating and representing users in public communication of research on Twitter

- I am missing any link or reference to topics such as “fake news”, “post facts”, etc.

- “Our study focuses on the analysis of perceived users as a proxy of the publics of research communication on Twitter.” (line162-163) – ok, but is this common standard? Please provide a context that underlines that this is an agreed upon, or suggested approach.

- “As such, we account for the multiple identity markers mobilized in individual description to further assess the complexity through which a user may engage with research documents.” (lines 185-187) – ok, but do you also consider the chronological order to markers? For example, a user could be mother, activist and researcher. This is complex. But do you account for mother first and then researcher? One could argue that the list is a meaningful choice by the person.

Purpose of the study

- “As such, it is useful to assess another user position on Twitter, especially when there is no known relationship.” (lines 191-192) – true. But how can this be done without social network analysis? Please elaborate.

- “As such, our study analyzes perceived users through the mobilization of specific keywords in Twitter descriptions.” (lines 194-195) – this is very vague to me. At this stage of the manuscript, I am still not sure what “perceived users” and “mobilization of specific keywords” really means. Please elaborate and explain exactly what this is.

Data collection and Twitter metrics

- “They also were the latest years for which we had complete Twitter information at the time of data collection in September 2018.” (lines 234-236) – what does this really mean? Complete Twitter information can be quite a lot of things. Moreover, the authors indicated before that Twitter data is “messy”. So what does this really entail?

- “It also frequently appears in tweets sharing a link to a paper, and so is highly visible to all users.” (lines 237-238) – highly visible is, I believe, an overstatement. If an individual researcher shares the link to her paper and uses a commonly used hashtag, her post will more than likely drown in the information overload. Please elaborate.

- “we computed several Twitter metrics to further describe our dataset for tweet activity.” (lines 259-260) – which metrics did you compute? It becomes apparent later on, but I think it should be stated here.

- Overall, the paper does not explain, to the best of my knowledge, how users’ profile data was collected. Please make sure that this is included and properly described.

Textual analysis of Twitter profile descriptions

- The first paragraph (lines 297-308) remains descriptive and does not include any references to previous research that has done similar work. Please rectify this.

- “we then improved it through several iterations of the analysis.” (lines 314-315) – how exactly did you do this?

Presenting our observations

- Again this section remains rather descriptive and does not really strike a link to previous research and studies. Please provide more information on how the research fits into the larger picture.

Discussion

- “To circumvent some of these issues, our analysis focused on a method to assess perceived users as publics involved in climate change research Twitter communication.” (lines 486-487) – After reading the manuscript and argumentation, I am unfortunately not convinced that the authors can really make such a statement. The Tweets were selected based on DOI. This neglects a wide range of hashtags that are commonly used in this space, where the public gets their information and where researchers “need to tap-in to”, in order to gain recognition. I also wonder about the content of the Tweets – which has been neglected. How did people engage?

Conclusion

- “While our results do not provide a direct assessment of who tweet research due to the characteristics of Twitter data, it provides insights about how documents may permeate outside of academia and in specific groups.” (lines 600-602) – while I agree with the second part of the statement, I tend to disagree with the first part. Social Network Analyses, among others, has been proven as a valuable tool to analyze Twitter communication streams. Hence, the “messy nature” of Twitter data cannot explain why this manuscript has not provided applicable information. Overall, this is my main issue with the manuscript. The authors remain descriptive on a wide range of key issues that would justify the chosen method framework. Moreover, some more insights are not elaborated on and merely mentioned. Finally, some statements are made based on very shaky ground, particularly in view of previous, interdisciplinary research that has been done on Twitter communication. I would like to encourage the authors to carefully reconsider their argumentation and justification, in order to enhance the quality of the manuscript, which otherwise provides an interesting approach to the field.

Reviewer #2: Overall, I think the paper is clever and uses some interesting methodology and data. However, I think that language might be the major issue here, since using words such as 'engagement' and 'mobilised' would imply a very different type of analysis. For example, answering Q1 would require an analysis of actual engagement: likes, retweets, comments, quotes etc.

The authors state a more realistic and appropriate goal for the paper on pg. 14: " Rather, we understand expressions in Twitter descriptions as a proxy to investigate the potential publics engaging with specific scholarly documents." But even here, the word "engaging" is problematic. Simply sharing/retweeting a research article is not necessarily indicative of engagement. Other problematic terms that are used in the manuscript are ‘resonance’, and 'publics of climate change’; these are undefined terms and their relevance is not argued for. ‘Distinct communicational contexts’ is another example of wording that seems vague or conflated. What is the meaning of ‘context’ in this manuscript?

The main comment that thus arises is: What is actually the (main) research objective? Who (re)tweets papers? (user focused) or how documents may permeate a social media environment? Both current research questions are formulated too ambitious if I take a critical look at the type of analyses and results. It is hard to understand how user characteristics translate into indications of ‘resonance’ (RQ1). RQ2 focuses on ‘implications’, but based on what results? This questions seems more suitable for the Discussion.

At best, the paper could make claims about assessing the types of users/publics that contribute to the spreading of climate change research or co-creating a -- research informed -- climate change narrative. In many ways, the manuscript seems to be more a methodological paper (i.e. presenting an innovative methodology) than actually presenting meaningful results.

Other points that need attention:

Besides the need for a stronger rationale for the study, the introduction seems to provide opportunities for a stronger structure. Descriptive part and concluding remarks mixed haphazardly. Some sections show overlap or repetition (e.g., scholars including hyperlinks).

Structure in Methods is not very clear. Notably, information about article dataset is scattered throughout this section, and description is mixed with Twitter dataset.

The structure in the Results can be structured more clearly, as the focus shifts from papers to users and vice versa.

The Results seem more top-down than suggested in Introduction; because there are 8 predefined categories. Was there room for a more explorative examination of users? Otherwise I suggest building a stronger argument for these categories, and earlier in the manuscript.

Conclusions can be based more explicitly on results. I find it difficult to see the merit of percentages of user categories related to article titles. It seems a big ‘leap’ to conclusions based on the kind of data.

Reviewer #3: The manuscripts presents methodology developed to automatically detect certain keywords in Twitter profiles in order to learn and categorize the Twitter users to predefined categories. The results are promising and the authors write that their methods and system could be adapted to investigate Twitter users in general or some other specific subgroup, besides those connected to climate change research. It is' however, clear whether the scripts have been made or will be made openly available for other researchers to test and use.

The results are sound and discussed in detail. The authors also identify some caveats in their work and discuss those accordingly. The term "perceived users" needs a bit more explaining. The paper has some minor issues with the use of prepositions and between singular/plural form of words. Please check the following lines for such (and some minor typos): 55, 134, 152, 176, 184, 186, 190, 191, 201, 227, 237, 264, 278, 280, 285 (reference form), 287, 305, 359 (reference form), 380, 408, 416, 460, 502.

6. PLOS authors have the option to publish the peer review history of their article (what does this mean?). If published, this will include your full peer review and any attached files.

Reviewer #1: No

Reviewer #2: **Yes: **Bob C Mulder

Reviewer #3: No

---

## [Author Response · Author response to Decision Letter 0]

5 Apr 2022

Dear Editors and Reviewers,

Please find below our replies to editors and referees’ comments (in red), as well as relevant line numbers associated with our modifications to the manuscript (for comments from the reviewers). 

Editor comments

1. Clear language, clearly defined terms and claims as pointed out by R#1, R#2 and R#3. Please either use different wordings or generally accepted definisions (backed up by proper references). If they do not exist, please make sure that they are defined in the manuscript.

We clarified the language and definitions when applicable. 

2. Please add missing references which would support the statements and assumptions in the manuscript, as well as provide a complete picture of previous research on the topic of the manuscript (R#1, R#2)

We added references to support our arguments as recommended by the reviewers.

3. Please ensure that conclusions are justified by the results (R#1, R#2)

We precised the conclusions according to the results.

4. Please ensure that the paper has a clear research objective (R#2)

We precised our objectives and research questions.

5. Please make sure that the data availability is in line with PLOS Data policy (R#3)

We verified the availability of our data according to the PLOS Data policy. 

We have changed to paper to match PLOS ONE’s format. We reformatted the title and section headings according to PLOS ONE’s style requirements.

2. In your Methods section, please include additional information about your dataset and ensure that you have included a statement specifying whether the collection method complied with the terms and conditions for the websites from which you have collected data.

We added a statement about the compliance of our data collection protocol according to all parties involved.

“We would like to thank Stefanie Haustein and Juan Pablo Alperin from the ScholCommLab for their help and feedback regarding the analysis. We would also like to thank Matisse Dagenais and Sandrine Dagenais in helping build the codebook. This research was funded through a SSHRC Joseph-Armand Bombardier Canada Graduate Scholarship (767-2017-1329), the SSHRC Insight Grant Understanding the societal impact of research through social media, and received financial contribution from the CIRST.”

“This study was funded by the Social Sciences and Humanities Research Council of Canada, the Fonds de recherche du Québec - Société et Culture, the Centre interuniversitaire de recherche sur la science et la technologie, the Université du Québec à Montréal and the Canada Research Chairs program. The funders had no role in study design, data collection and analysis, decision to publish, or preparation of the manuscript.”

We removed funding information from the Acknowledgements and added instructions to update to our Funding Statement to the cover letter.

We added captions for the Supporting Information files and updated in-text citations according to PLOS ONE’s guidelines.

REVIEWERS' COMMENTS:

Reviewer #1: 

- Please elaborate on what you mean by “messy data from Twitter” (lines 53-55)

We clarified this part of the sentence, we meant that Twitter data is mostly generated by users and readily organized for research purposes.

- “As such, science communication and policy increasingly aim to understand the resonance of research in the public sphere.” (lines 58-59) – Really? Please provide more information on this and elaborate why you think this is the case. More specifically, policy has often been criticized for detaching itself from research and rather follow “popular opinion”.

We clarified this sentence. We meant that studies and policy documents about the communication and evaluation of research put an increasing focus on the impact of research in the public sphere for evaluation pruposes.

- The cited articles (lines 71-82) seem to fit the argument. However, more information is needed on the specific studies, in order to better understand and contextualize their importance for this manuscript.

We added information about the cited articles in order to contextualize their contribution to the manuscript.

- “While patterns of scholarly communication on Twitter remain to be documented in detail, it is considered to have democratized access to research.” (lines 91-92) – this is a very bold statement and should be supported by further references. More specifically, one could argue that OER and OpenAccess publications have contributed more to this suggested process than Twitter.

We clarified this sentence. Mostly, we want to highlight that there are promises that sharing research on social media is democratizing research, independantly of the degree to which this may or not be the main factor. 

- The authors refer to Haustein, as well as Diaz-Faes – this needs more information and elaboration as it appears to be key to the manuscript.

We restructured the paragraph (line 95-117) to put the focus on the content as a whole (why we should investigate the contexts of research communication on Ywitter), and not strictly on Haustein and Diaz-Faes.

- What exactly are “non-elite” actors? (line 121)

We added a precision about what are “non-elite” actors according to Newman’s study.

Investigating and representing users in public communication of research on Twitter

- I am missing any link or reference to topics such as “fake news”, “post facts”, etc.

We did not include references to fake news or post facts as it was not directly related to the topic of the paper. While it is true that these are key issues for science communication on Twitter (and social media in general), this section aimed to provide an overview of the methods used so far to investigate who share research on Twitter.

- “Our study focuses on the analysis of perceived users as a proxy of the publics of research communication on Twitter.” (line162-163) – ok, but is this common standard? Please provide a context that underlines that this is an agreed upon, or suggested approach.

We added some more information to this. The idea is that investigating who tweet research paper usually entails reducing identity to specific markers, such as this is a scientist or not.

- “As such, we account for the multiple identity markers mobilized in individual description to further assess the complexity through which a user may engage with research documents.” (lines 185-187) – ok, but do you also consider the chronological order to markers? For example, a user could be mother, activist and researcher. This is complex. But do you account for mother first and then researcher? One could argue that the list is a meaningful choice by the person.

We did not took the chronological order into account as we wanted to avoid presuming which identities are more important. Basically, our method relies on asking questions such as “does this user present itself through keywords related to academia or not?”

Purpose of the study

- “As such, it is useful to assess another user position on Twitter, especially when there is no known relationship.” (lines 191-192) – true. But how can this be done without social network analysis? Please elaborate.

We changed “position” for “orientation or interest” to highlight that Twitter bios may provide cues about the inclination toward specific topics, instead of a position from a network perspective. Some of our future work about this will focus on social network analysis.

- “As such, our study analyzes perceived users through the mobilization of specific keywords in Twitter descriptions.” (lines 194-195) – this is very vague to me. At this stage of the manuscript, I am still not sure what “perceived users” and “mobilization of specific keywords” really means. Please elaborate and explain exactly what this is.

We removed mention of “perceived users” as we felt it did not improve the paper. We also clarified our objective.

Data collection and Twitter metrics

- “They also were the latest years for which we had complete Twitter information at the time of data collection in September 2018.” (lines 234-236) – what does this really mean? Complete Twitter information can be quite a lot of things. Moreover, the authors indicated before that Twitter data is “messy”. So what does this really entail?

We clarified this statement. We meant that they were the most recent years for which we had altmetric information for all papers published in both years.

- “It also frequently appears in tweets sharing a link to a paper, and so is highly visible to all users.” (lines 237-238) – highly visible is, I believe, an overstatement. If an individual researcher shares the link to her paper and uses a commonly used hashtag, her post will more than likely drown in the information overload. Please elaborate.

We clarified this sentence to highlight that it is usually the most visible information in tweets sharing research articles.

- “we computed several Twitter metrics to further describe our dataset for tweet activity.” (lines 259-260) – which metrics did you compute? It becomes apparent later on, but I think it should be stated here.

The metrics are listed in the following lines (lines 252-259 of the revised manuscript)

- Overall, the paper does not explain, to the best of my knowledge, how users’ profile data was collected. Please make sure that this is included and properly described.

We added information about this at line 220 of the revised manuscript.

Textual analysis of Twitter profile descriptions

- The first paragraph (lines 297-308) remains descriptive and does not include any references to previous research that has done similar work. Please rectify this.

We cut the first part of the paragraph which was redundant with elements mentionned in the introduction and merged the remaining part with the second paragraph.

- “we then improved it through several iterations of the analysis.” (lines 314-315) – how exactly did you do this?

We added some elements about the iterating process of the analysis and how we built the codebook.

Presenting our observations

- Again this section remains rather descriptive and does not really strike a link to previous research and studies. Please provide more information on how the research fits into the larger picture.

We removed this section as it did not add critical information to the manuscript.

Discussion

- “To circumvent some of these issues, our analysis focused on a method to assess perceived users as publics involved in climate change research Twitter communication.” (lines 486-487) – After reading the manuscript and argumentation, I am unfortunately not convinced that the authors can really make such a statement. The Tweets were selected based on DOI. This neglects a wide range of hashtags that are commonly used in this space, where the public gets their information and where researchers “need to tap-in to”, in order to gain recognition. I also wonder about the content of the Tweets – which has been neglected. How did people engage?

We rephrased this statement to precise that we focus on a method to categorized who share research articles on Twitter. Our units are the tweets that include a link to a climate change research paper, so we didn’t include the hashtags in our analysis as it wasn’t the focus of our study. We also didn’t look at the content of the tweets for now, but we plan to for a subsequent study.

Conclusion

- “While our results do not provide a direct assessment of who tweet research due to the characteristics of Twitter data, it provides insights about how documents may permeate outside of academia and in specific groups.” (lines 600-602) – while I agree with the second part of the statement, I tend to disagree with the first part. Social Network Analyses, among others, has been proven as a valuable tool to analyze Twitter communication streams. Hence, the “messy nature” of Twitter data cannot explain why this manuscript has not provided applicable information. Overall, this is my main issue with the manuscript. The authors remain descriptive on a wide range of key issues that would justify the chosen method framework. Moreover, some more insights are not elaborated on and merely mentioned. Finally, some statements are made based on very shaky ground, particularly in view of previous, interdisciplinary research that has been done on Twitter communication. I would like to encourage the authors to carefully reconsider their argumentation and justification, in order to enhance the quality of the manuscript, which otherwise provides an interesting approach to the field.

We added references from interdisciplinary research about scholarly communication on Twitter and refraned the paper to make our argument more clear. Our main objective is to provide a proxy and some new groundwork about ‘who’ tweets climate change research papers for future studies about the communication of science on Twitter in particular and digital media in general. Our approach can also be used in conjonction with other methods to improve what we know about the diffusion of research by providing a quick and flexible typology of users based on what they chose to put in their Twitter profile descriptions. This typology can be then used in Social Network Analyses to categorize the nodes that shared research papers on Twitter, as an example.

Reviewer #2:

Overall, I think the paper is clever and uses some interesting methodology and data. However, I think that language might be the major issue here, since using words such as 'engagement' and 'mobilised' would imply a very different type of analysis. For example, answering Q1 would require an analysis of actual engagement: likes, retweets, comments, quotes etc.

We rectified the language and clarified our study in regard to these considerations.

The authors state a more realistic and appropriate goal for the paper on pg. 14: " Rather, we understand expressions in Twitter descriptions as a proxy to investigate the potential publics engaging with specific scholarly documents." But even here, the word "engaging" is problematic. Simply sharing/retweeting a research article is not necessarily indicative of engagement. Other problematic terms that are used in the manuscript are ‘resonance’, and 'publics of climate change’; these are undefined terms and their relevance is not argued for. ‘Distinct communicational contexts’ is another example of wording that seems vague or conflated. What is the meaning of ‘context’ in this manuscript?

We revised wording through the manuscript. We added a definition to “context” in regard to our study : Scholars thus looked to investigate the contexts, understood as the dimensions that give meaning to indicators, in which research circulate on Twitter (4). (p.3)

The main comment that thus arises is: What is actually the (main) research objective? Who (re)tweets papers? (user focused) or how documents may permeate a social media environment? Both current research questions are formulated too ambitious if I take a critical look at the type of analyses and results. It is hard to understand how user characteristics translate into indications of ‘resonance’ (RQ1). RQ2 focuses on ‘implications’, but based on what results? This questions seems more suitable for the Discussion.

We rephrased RQ1 as our main research question (RQ) : How do climate change research papers, both individually and as a whole, get shared outside of academia? (p. 10)

We also removed QR2 as it was not central to the manuscript and added information about the implications and limitations of our method in the Discussion.

At best, the paper could make claims about assessing the types of users/publics that contribute to the spreading of climate change research or co-creating a -- research informed -- climate change narrative. In many ways, the manuscript seems to be more a methodological paper (i.e. presenting an innovative methodology) than actually presenting meaningful results.

We revised the manuscript according to a more precise goal : assess how much climate change research papers are being shared outside of academia. 

Other points that need attention:

Besides the need for a stronger rationale for the study, the introduction seems to provide opportunities for a stronger structure. Descriptive part and concluding remarks mixed haphazardly. Some sections show overlap or repetition (e.g., scholars including hyperlinks).

We revised the introduction and clarified its structure. We also revised the manuscript to minimize overlap when applicable.

Structure in Methods is not very clear. Notably, information about article dataset is scattered throughout this section, and description is mixed with Twitter dataset.

We revised the structure of the Methods section to make it clearer. 

The structure in the Results can be structured more clearly, as the focus shifts from papers to users and vice versa.

We revised the structure of the Results to make them clearer. We however kept the general structure of general results first, then results for highly tweeted papers for the categories Academia, Communication, Political, Professional, and then Personal.

The Results seem more top-down than suggested in Introduction; because there are 8 predefined categories. Was there room for a more explorative examination of users? Otherwise I suggest building a stronger argument for these categories, and earlier in the manuscript.

We added information about the chosen categories in the litterature review. 

Conclusions can be based more explicitly on results. I find it difficult to see the merit of percentages of user categories related to article titles. It seems a big ‘leap’ to conclusions based on the kind of data.

We revised the conclusions according to the results.

Reviewer #3: 

The manuscripts presents methodology developed to automatically detect certain keywords in Twitter profiles in order to learn and categorize the Twitter users to predefined categories. The results are promising and the authors write that their methods and system could be adapted to investigate Twitter users in general or some other specific subgroup, besides those connected to climate change research. It is' however, clear whether the scripts have been made or will be made openly available for other researchers to test and use.

We uploaded the scripts to Github for further improvement of the method and dictionnaries.

The results are sound and discussed in detail. The authors also identify some caveats in their work and discuss those accordingly. The term "perceived users" needs a bit more explaining. 

We removed mentions of the term “perceived users” as it was confusing and not useful pour the interpretation of the paper. We focused instead on the categories of users we can assess by examining Twitter profile descriptions.

The paper has some minor issues with the use of prepositions and between singular/plural form of words. Please check the following lines for such (and some minor typos): 55, 134, 152, 176, 184, 186, 190, 191, 201, 227, 237, 264, 278, 280, 285 (reference form), 287, 305, 359 (reference form), 380, 408, 416, 460, 502.

We corrected the wording, issues and typos in the above-mentionned lines.

---

## [Decision Letter · Decision Letter 1]

13 May 2022

Who tweets climate change papers? Investigating publics of research through users’ descriptions

PONE-D-21-17099R1

Dear Dr. Toupin,

We’re pleased to inform you that your manuscript has been judged scientifically suitable for publication and will be formally accepted for publication once it meets all outstanding technical requirements.

Kind regards,

Piotr Bródka

Academic Editor

PLOS ONE

---

## [Editor Report · Acceptance letter]

23 May 2022

PONE-D-21-17099R1 

Who tweets climate change papers? Investigating publics of research through users’ descriptions 

Dear Dr. Toupin:

I'm pleased to inform you that your manuscript has been deemed suitable for publication in PLOS ONE. Congratulations! Your manuscript is now with our production department. 

Kind regards, 

on behalf of

Dr. Piotr Bródka 

Academic Editor

PLOS ONE